# Thermodynamic Analysis of a Hybrid Trigenerative Compressed Air Energy Storage System with Solar Thermal Energy

**DOI:** 10.3390/e22070764

**Published:** 2020-07-13

**Authors:** Xiaotao Chen, Xiaodai Xue, Yang Si, Chengkui Liu, Laijun Chen, Yongqing Guo, Shengwei Mei

**Affiliations:** 1Qinghai Key Lab of Efficient Utilization of Clean Energy (New Energy Photovoltaic Industry Research Center), Qinghai University, Xining 810016, China; chenxiaotao@qhu.edu.cn (X.C.); lurkaries@gmail.com (Y.S.); yqguo2013@163.com (Y.G.); meishengwei@tsinghua.edu.cn (S.M.); 2China State Key Laboratory of Power System and Generation Equipment, Department of Electrical Engineering, Tsinghua University, Beijing 100084, China; xuexiaodai@mail.tsinghua.edu.cn; 3Qinghai Building and Materials Research Co, Ltd., Xining 810008, China; qhklpbec@163.com; 4The Key Lab of Plateau Building and Eco-community in Qinghai, Xining 810008, China

**Keywords:** hybrid T-CAES, solar energy, solar adsorption chiller, performance analysis, HTF ratio for heating and cooling

## Abstract

The comprehensive utilization technology of combined cooling, heating and power (CCHP) systems is the leading edge of renewable and sustainable energy research. In this paper, we propose a novel CCHP system based on a hybrid trigenerative compressed air energy storage system (HT-CAES), which can meet various forms of energy demand. A comprehensive thermodynamic model of the HT-CAES has been carried out, and a thermodynamic performance analysis with energy and exergy methods has been done. Furthermore, a sensitivity analysis and assessment capacity for CHP is investigated by the critical parameters effected on the performance of the HT-CAES. The results indicate that round-trip efficiency, electricity storage efficiency, and exergy efficiency can reach 73%, 53.6%, and 50.6%, respectively. Therefore, the system proposed in this paper has high efficiency and flexibility to jointly supply multiple energy to meet demands, so it has broad prospects in regions with abundant solar energy resource.

## 1. Introduction

Nowadays, the balance between cooling, heating and power supply and demand has become a key issue in many countries, due to the increasing penetration of intermittent renewable energy source (RES) and distributed generation (DG) [1,2]. The RES, such as wind and sun, are greatly subject to local environmental conditions and unpredictable weather, which will cause inconvenience for energy utilization. Besides, the inherent intermittency of renewable DG will also affect the reliability, efficiency and safety of the distributed energy system (DES) [3]. Therefore, exploring a safe, reliable, efficient, and economical DES which combines cooling, heating and power, has been the focus in the DES research field.

Presently, the combined cooling, heating and power (CCHP) system and integrated with the energy storage system are the major multi-carrier energy hub technologies to cope with this problem [4]. It would serve to store the low quality (fluctuating and intermittent) energy, and provide high quality (smooth) and dispatchable cooling heating and power (CHP) based on consumption needs [5]. Among various energy storage and CCHP technologies, compressed air energy storage (CAES) may act as trigenerative systems (T-CAES), which can meet the various forms energy demand of users by recovering and supplying heating by stored compressed heat and cooling energy during the expansion [6]. Three types of CAES systems have been extensively investigated in the literature: diabatic (D-CAES), isothermal (I-CAES), and adiabatic (A-CAES) [7,8,9,10]. Only two existing D-CAES plants have been constructed until now: the Huntorf plant of 290 MW, Germany, was constructed in 1978; it is worth mentioning that this plant was expanded to 321 MW in 2007. The McIntosh of 110 MW, USA, was built in 1991. However, D-CAES relies on natural gas as an external heat source which can cause certain carbon emissions [11,12]. For these reasons, significant efforts have been devoted to I-CAES and A-CAES, which address the drawbacks of D-CAES by avoiding the use of fossil fuels and recycle compression heat for enhancing their performance.

Recently, most of the literature related to A-CAES is focused on utility-scale applications [13,14,15], which are large-scale power plants located close to energy production. Nevertheless, a small-scale A-CAES (micro-CAES) may act as a trigenerative CAES (T-CAES) that is placed close to the user side and possess a tighter coupling between the DG and the energy demand, has received great attention in DES [16,17]. Andrea L. Facci et al. [18] introduce the concept of a trigenerative compressed air storage (T-CAES), which is based on a micro-CAES. However, constrained by the limited capacity and effectiveness of the heat transfer in the micro-CAES, thermal energy storage (TES) that only depended on compression heat cannot reach a high temperature, which restricted the performance and energy density of micro-CAES systems [19,20,21].

Apart from micro-CAES acting as CCHP, another promising solution for improving the performance of the above system is hybrid external solar thermal energy and CAES (ST-CAES), which could notably enhance the storage heat temperature and capacity [22]. Besides, as a high-performance CCHP, ST-CAES can also enhance the comprehensive utilization level of clean energy [23,24,25,26,27]. Therefore, scholars and engineers have recently focused on the essential technologies of HT-CAES, such as process design, efficiency analysis, and key parameter optimization, for improving its comprehensive energy utilization efficiency.

Chen et al. [22] presented a novel solar thermal-assisted A-CAES to attain stable high-grade thermal energy and analyzed its thermodynamic characteristic. Xu et al. [23] proposed a novel scheme of a wind energy complementary A-CAES and analyzed its thermodynamic performance. Ji et al. [28] proposed a gas turbine combined solar energy and CAES system and focused on process design and efficiency analysis. Mohammadi et al. [29] analyzed an integrated micro gas turbine, compressed air energy storage, and solar dish collector system. Semprini et al. [30] investigated a hybrid-ACAES (HA-CAES) design scheme to optimize cycle efficiency and efficiency for microturbines and solar dish collectors. Yang et al. [31] analyzed the CCHP integrated solar thermal energy for its cogeneration characteristics, and better energy-saving and CO_2_ reduction performance is obtained. Li et al. [32] proposed an HA-CAES based multi-carrier clean energy hub to realize the comprehensive consumption of renewable energy. Mei et al. [33] studied an HA-CAES as a clean energy hub in a smart micro energy grid based on solar energy.

The previous studies for the HA-CAES mainly focused on the theoretical analysis and simulation of the system design and optimization. Few studies have worked on assessing the efficiency and capacity of T-CAES with stable solar thermal resources for providing CHP. As we know, no public analysis data have been elaborated on the CHP performance of the HA-CAES system. In this research, a hybrid solar thermal trigenerative CAES is proposed and thermodynamic analysis for its efficiency and capacity of CHP has been carried out.

The major contributions of this paper are: (1) A novel hybrid T-CAES with a solar thermal energy collection field and absorption chiller for supplying CHP was presented. (2) The HT-CAES using VP-1 as the thermal energy storage working medium was built and the thermodynamic performance and CHP capacity were investigated. (3) HT-CAES can provide stable high-grade solar thermal energy in the discharging process, compared with A-CAES in [10], which can greatly simplify the regenerative system and enhance its efficiency and flexibility.

The rest of paper is organized as follows. Section 2 describes the overall design process and thermodynamic model of HT-CAES. Section 3 elaborates the energy and exergy analysis of HT-CAES. This is greatly related to the critical parameters and the impact of the CHP capacity assessment on system performance in Section 4, followed by conclusions in Section 5.

## 2. System Description

Figure 1 shows the schematic diagram of the proposed HT-CAES system. To clearly describe and analyze the system, all the components and streams have been numbered. The bottom part of Figure 1 is a small-scale AA-CAES system where compression heat generated in the charging process is stored in the high-temperature water tank (HWT) to provide heat for residents.

An HT-CAES is mainly composed of five units, i.e., compression air storage unit (COM), air turbine and generator unit (TUR), solar thermal collecting and storage unit (STS), and solar absorption chiller unit (SAC). The COM unit contains a motor (M), air compressor train (AC1–AC4), four heat exchangers (HE1–HE4) in series, a hot-water tank (HWT), a low-temperature water tank (LWT), a compression heat radiator (CHR) and an air storage chamber (ASC). The TUR unit includes a throttle valve (TV), an air turbine chain (AT1, AT2 and AT3), a preheat regenerator (PHR) and three heat exchangers (HE5, HE6 and HE7). The STS unit consists of a parabolic trough collector (PTC), a cold oil tank (COT), a hot oil tank (HOT), and heat transfer fluid (HTF). The HTF used in the absorber tube was Therminol^®^VP-1, which is widely used in STS system [34]. It has an optimal usage range when in liquid phase from 12 °C to 400 °C [35]. Therefore, Therminol^®^VP-1 is quite suitable for the proposed HT-CAES. The SAC unit comprises refrigeration air-conditioning (RAC), a chiller machine (CM), a cooling water tank (CWT) and LWT.

Although the COM and TUR subsystems are greatly similar to the A-CAES reported in many studies [10,11,12,13,14,15], the HT-CAES does possess several distinct characteristics. Firstly, compression heat energy is used to supply heating to users. Secondly, high-temperature VP-1 heated by the PTC can supply sufficient and high-grade stable heat resources for SAC; VP-1 in HOT and compressed air can provide cooling, heating, and power. Thirdly, the compressed air is preheated by the exhaust gas of the last stage of the expander before entering the turbo expander and is further heated by the high-temperature VP-1 of the heat exchanger (HE5–HE7) to enhance the comprehensive efficiency of HT-CAES.

## 3. Thermodynamic Analysis Model

To analyze the performance of the hybrid T-CAES, a comprehensive thermodynamic model is established in this section based on mass and energy balance. The classical Peng–Robinson equations of the state were selected as the property package. The analysis was carried out using Thermoflow software. To facilitate the analysis, some assumptions are made as follows:Potential and kinetic energies of all units are negligible;All gases in the system are treated as the ideal gas, and the Joule–Thomson effect is negligible;Compression in the compressor and expansion in the turbine is regarded as an isentropic process;In the charging or discharging process, the air storage chamber is considered as an isothermal process;Compressed air can be stored at constant volume, and its temperature in ASC is the same as ambient temperature;The pressure loss in all pipes and heat exchangers is ignored.

### 3.1. STS

The STS includes PTC and a thermal storage tank. The PTC thermodynamic model is shown in [22]. The thermal storage tanks are modeled by dynamic mass and energy balances for two tanks. The mass balance for a tank is:(1)$$\rho_{HTF}\frac{dV_{HTF}}{dt}={\dot{m}}_{in}-{\dot{m}}_{out},$$
where *V**_HTF_* is the volume of heat transfer fluid (HTF) in the tank, and *ρ****_HTF_*** refers to the density of HTF.

The energy balance for each tank is:(2)$$\rho_{HTF}C_{HTF}\frac{d(V_{HTF}T)}{dt}=C_{HTF}\left(T_{in}{\dot{m}}_{in}-T_{in}{\dot{m}}_{out}\right)-UA_{t}\left(T-T_{0}\right),$$
where U is the overall heat transfer coefficient for the tank walls and *A_t_* is the surface area of the tank subject to the heat transfer. We assume that no heat transfer occurs from the top or bottom of either tank because the volumes of HTF (*V_HTF_*) in the tanks are not constant.

### 3.2. SAC

The solar absorption chiller is driven by the collected heat of STS. The heat of Therminol oil VP-1 (VP-1) supplied to the vapor generator of the absorption chiller (*Q_ab_*(*t*)) can be written as:(3)$${\dot{Q}}_{ab}\left(t\right)=c_{p,O}{\dot{m}}_{O,ab}\left(t\right)\left(T_{O13}-T_{O14}\right),$$
where ${\dot{m}}_{O,ab}\left(t\right)$ is the mass flow rate of hot oil (VP-1) injecting into the absorption chiller. $T_{O13}$ is the temperature of stream O13. $T_{O14}$ is the temperature of stream O14. The cooling load production by the absorption chiller (${\dot{Q}}_{cl,ab}\left(t\right)$) can be obtained by,
(4)$${\dot{Q}}_{cl,ab}\left(t\right)=COP_{ab}\cdot {\dot{Q}}_{ab}\left(t\right),$$
where $COP_{ab}$ is the coefficient of performance of the absorption chiller.

### 3.3. T-CAES

#### 3.3.1. Charging Process

During the charging process, the power produced by PV or wind can be used to drive the compressor of T-CAES.
(5)$${\dot{W}}_{COM}\left(t\right)=\sum_{i=1}^{4}{\dot{m}}_{ac,T-CAES}\left(h_{out}^{i}-h_{in}^{i}\right),$$
where ${\dot{W}}_{COM}\left(t\right)$ is the compressor power consumption and ${\dot{m}}_{ac,T-CAES}\;$is the mass flow rate of the compressor, which is produced by wind or solar energy.$\;h_{in}^{i}$ and $h_{out}^{i}$ are the compressor inlet and outlet enthalpy of each stage.

In the compression process, the parameters of stream A1 are the same as the ambient, and the parameters of air outlet the compressor (stream A2, A4, A6 and A8) are described as follows:(6)$$p_{COM,i}=\beta \cdot p_{0},$$
(7)$$T_{i}=\frac{T_{0}\left(\beta^{\frac{\kappa -1}{\kappa }}-1\right)}{\eta_{c,T-CAES}}+T_{0},$$
where β is the pressure ratio of the compressor in the T-CAES. $T_{0}$ and $p_{0}\;$are the ambient temperature and pressure. $\eta_{c,T-CAES}\;$is the isentropic efficiency of the compressor in the T-CAES and κ is the polytropic index.

The air mass flow rate of the compressor in the T-CAES can be calculated by:(8)$${\dot{m}}_{ac,T-CAES}\left(t\right)=\frac{{\dot{W}}_{COM}\left(t\right)}{c_{p,a}\left(T_{6}-T_{0}\right)},$$
where $c_{p,a}$ is the air-specific heat at constant pressure.

The high-pressure air is cooled down through the side of the COM heat exchanger. The heat recovered in the charging process (${\dot{Q}}_{cp,HT-AES}\left(t\right)$) and can be written as:(9)$${\dot{Q}}_{cp,T-AES}\left(t\right)=\sum_{i=2}^{9}c_{p,a}{\dot{m}}_{ac,T-CAES}\left(t\right)\cdot \left(T_{Ai}-T_{Ai+1}\right),$$

After cooling, the stream A9 flows into the air storage chamber (ASC). The temperature of the compressed air storage tank stays the same as the ambient temperature *T*_0_. The final air mass and pressure of compressed air inside ASC are determined as follows:(10)$$m_{0,ASC}=\frac{p_{0,ASC}V_{ASC}}{R_{a}T_{0}},$$
(11)$$p_{d,ASC}=\frac{\left(m_{0,ASC}+\int_{0}^{\tau_{ch}}{\dot{m}}_{ac,HT-CAES}\left(t\right)dt\right)R_{a}T_{0}}{V_{ASC}},$$
where $\tau_{ch}$ is the charging time, $m_{0,ASC}\;$is the initial air mass of ASC before the charging process, and $p_{0,ASC}$ is the initial air pressure of the compressed air chamber before charging process. $p_{d,ASC}$ is the final pressure of the compressed air chamber after charging process. $V_{ASC}$ is the volume of the compressed air reservoir. *Ra* is the air gas constant.

#### 3.3.2. Discharging Process

During the discharging process, the volume of compressed air chamber keeps constant. As the discharging process goes on, the inside pressure of compressed air chamber drops. The temperature of stream A10 is the same as ambient temperature *T*_0_, and the pressure of ASC (stream A10) can be calculated as follows:(12)$$p_{A10}=\frac{\left(m_{0,ASC}+\int_{0}^{\tau_{dch}}{\dot{m}}_{at,HT-CAES}\left(t\right)dt\right)R_{a}T_{0}}{V_{ASC}},$$
(13)$$\{\begin{array}{c} 0\leq t\leq \tau_{dch}\\ 0\leq \int_{0}^{\tau_{dch}}{\dot{m}}_{at,T-CAES}\left(t\right)dt)\leq m_{0,ASC}+\int_{0}^{\tau_{ch}}{\dot{m}}_{ac,T-CAES}\left(t\right)dt) \end{array}$$
where ${\dot{m}}_{ac,T-CAES}$ is the air mass flow rate of turbine in the TUR. $\tau_{dch}$ is the discharging time. This research only considers the regulation of air flow rate by adjusting the regulating valve and the Joule–Thomson effect is negligible. Before entering each stage of turbine, the stream A11 is first preheated by the third stage expander, and then, heated by stored solar thermal energy in the HOT. The temperature of stream A13 can be calculated according to energy conversion equation, and the pressure of stream A13 equals to the pressure of stream A11 and A12.
(14)$$T_{A13}=\frac{c_{p,O}{\dot{m}}_{O}\left(t\right)\left(T_{O6}-T_{O7}\right)}{c_{p,a}{\dot{m}}_{at,HT-CAES}\left(t\right)}+T_{A12},$$
where ${\dot{m}}_{O}\left(t\right)$ is the mass flow rate of VP-1 using to heat stream A12. *T_O_*_6_ and *T_O_*_7_ are the temperatures of stream O6 and O7, respectively. The turbine outlet air temperature in the expansion process can be written as follows:(15)$$\pi_{i,TUR}=\frac{P_{TUR,i}^{in}}{P_{TUR,i}^{out}},$$
(16)$$T_{TUR}^{out}=T_{TUR}^{in}-\eta_{en,i}T_{TUR}^{in}\left(1-\pi_{i,TUR}^{\frac{1-\kappa }{\kappa }}\right)$$
where $\pi_{i,TUR}$ is the pressure ratio of the turbine. $\eta_{en,i}$ is the isentropic efficiency of each stage turbine in the T-CAES. The output power of the turbine in the T-CAES is presented as follows:(17)$${\dot{W}}_{TUR}\left(t\right)=\sum_{j=1}^{3}{\dot{m}}_{at,T-CAES}\left(t\right)\left(h_{TUR}^{out,i}-h_{TUR}^{in,i}\right)$$

### 3.4. Exergy Analysis Model

An exergy analysis based on the second law of thermodynamics can be performed to decide the exergy destruction of each subsystem [36]. Generally, the enthalpy exergy of the state *i* can be calculated by
(18)$$Ex_{i}={\dot{m}}_{i}\left[\left(h_{i}-h_{0}\right)-T_{0}\left(s_{i}-s_{o}\right)\right],$$
where ${\dot{m}}_{i}$ is the mass flow rate, h is the specific enthalpy, *s* is the specific entropy, and subscripts *i* and 0 represent state *i* and ambient conditions, respectively.

For each subsystem *j*, the exergy destruction and exergy efficiency can be calculated as follows:(19)$$L_{j}=E_{j,in}-E_{j,out},$$
(20)$$\eta_{EXE,j}=\frac{E_{j,out}}{E_{j,in}},$$
where the subscripts in and out represent the input and output states, respectively, of subsystem *j*.

The expressions for input and output exergy of each subsystem are listed in Table 1. It is worth noting that the abbreviations in Table 1 are list in the part of Nomenclature.

### 3.5. Performance Criteria

The proposed HT-CAES system has three modes of operation: energy storage, idle, and energy release mode. Unlike conventional CAES, energy storage modes can be divided into air storage and solar thermal collection and storage processes. It is worth noting that the process of collecting and storing solar thermal energy is independent of the air compression process. Therefore, these two processes can be performed simultaneously. In an idle mode, high-pressure air is stored in the ASC at the design pressure, and high-temperature VP-1 is stored in the HOT at the design temperature. In the energy release mode, the high-pressure air and the thermal energy are released at the same time. The compressed air is heated by the high-temperature VP-1 in the HX5–HX7 to drive the turbine to generate electrical energy. Electric storage efficiency (ESE), round-trip efficiency (RTE), and exergy efficiency (EXE) are key indicators for analyzing the performance of the proposed system. ESE is defined as the amount of electricity generated during discharging divided by the power consumption during charging. The power of the water pump and oil pump is ignored because of their low value. Therefore, ESE can be expressed as
(21)$$\eta_{ESE}=\frac{{\dot{W}}_{TUR}\cdot \tau_{dch}}{{\dot{W}}_{COM}\cdot \tau_{ch}}.$$

RTE is defined as the ratio of total thermal and electrical energy output to total solar and electrical energy input in a full charge/discharge cycle. It can be expressed as
(22)$$\eta_{RTE}=\frac{W_{TUR}\cdot \tau_{dch}+Q_{heat}\cdot \tau_{hs}+Q_{CS}\cdot \tau_{cs}}{W_{COM}\cdot \tau_{ch}+Q_{u}\cdot \tau_{c}}.$$

The exergy efficiency can be represented as
(23)$$\eta_{EXE}=\frac{W_{TUR}\cdot \tau_{dch}+Ex_{heat}\cdot \tau_{hs}+Ex_{CS}\cdot \tau_{cs}}{W_{COM}\cdot \tau_{ch}+Ex_{HTF}\cdot \tau_{c}},$$
where $Q_{heat}$ is the thermal energy supply to the heat load, $Q_{CS}$ is the cooling energy that generated by absorption chiller supply to the residents collected and stored solar thermal energy, $Ex_{heat}$ is enthalpy exergy of $Q_{heat}$, $Ex_{HTF}$ is the HTF enthalpy exergy absorbed by the PTC, and $Ex_{CS}$ is enthalpy exergy of $Q_{CS}$.

## 4. Results and Discussion

In this section, the performance of the hybrid T-CAES under typical operational conditions is analyzed and discussed. Table 2 lists the design parameters of the system. In this case, the duration of charging and discharging time was 5 h and 1.4 h, respectively. The mass flow rate of the compression and expansion process was 0.33 kg/s and 1.17 kg/s. The isentropic efficiency of the air compressors was 90%. The pressure of the outlet air of the throating valve (state A11), which is determined by the expansion ratio of each stage. The minimum ASC pressure is approximately equal to the pressure of the inlet air of the first-stage air turbine (state A13). From the designed parameters presented in this section, the compressor outlet pressure and expander inlet pressures were selected as 8 MPa and 3 MPa. The effects of the compression and expansion pressures on system performance are discussed in Section 4.2. In the TUR subsystem, the isentropic efficiency of the turbines was selected at 85% and the other parameters of the STS were selected by previous studies.

### 4.1. Typical Operational Conditions

Table 3 lists the main simulation results of the HT-CAES system under the designed conditions. Table 4 and Table 5 list the stream thermodynamic parameters of air, water, and Therminol VP-1 respectively. As shown in Table 3, the total compressed power consumption was 190 kW and the heat energy collected from solar was 100.6 kW. The output power of the turbine was 352 kW. The ESE, RTE, and exergy efficiency of the proposed system were 53.6%, 73%, and 50.6%, respectively. As listed in Table 4, the temperatures of supply and return water for the heating load were 60 °C and 20 °C. Under average solar irradiation operational conditions, the system needed about 6 h to raise 7.2 tons of VP-1 from 100 °C to 250 °C. The output power of the turbine is greatly influenced by the VP-1 temperature provided by the collected and stored solar energy, which is discussed in Section 4.2. The performance of the A-CAES depends on the regenerative system, which is to recover and store compression heat in thermal energy storage (TES). This is greatly constrained by the structure of the compressor and multi-stage heat exchanger effectiveness. The adopted solar thermal energy can avoid the high-temperature limit of the compressor and complex heat regeneration subsystem, which can greatly simplify the structure of the A-CAES [22]. For a 4 h discharge, the total power generation capacity was 955.4 kWh.

To evaluate the performance of each subsystem, the efficiency and exergy destruction were calculated from Equations (19) and (20) and Table 1. Figure 2 illustrated the exergy efficiency of each subsystem. Due to the basic characteristics of STS, the exergy efficiency of the STS subsystem is much lower than that of other subsystems, except TUR’s HE. In the STS, the absorption energy of the heat medium was provided by stored Therminol VP-1 in the HOT, which was greatly affected by the PTC’s mirror field area, mass flow rate, weather, optical efficiency, and tracking accuracy [37]. In addition, due to the large temperature difference between cold and hot fluids, the HE of TUR exergy efficiency is the lowest in the discharging process.

Figure 3 shows the exergy destruction of each subsystem. In the STS system, the exergy destruction is 14% because of the irreversible loss caused by the external heat conduction, convection and the radiation of the parabolic mirror and absorber tube surface [38,39]. The STS requires a long time to heat the VP-1 from ambient temperature to the design temperature before the operation. In this process, the exergy destruction of the system does not take into account the exergy analysis of the HT-CAES. Due to the high temperature differential heat transfer, the exergy destruction of AT in TUR is the largest of all subsystems, which results in irreversible heat loss.

To further clarify the performance of the proposed HT-CAES, a comparison of the HT-CAES with the HA-CAES in [29] was illustrated in Table 6. In the comparative case, the RTE and EXE of each system were 73% and 76.5%, and 50.6% and 53.4%, respectively. The inlet temperature of the gas turbine was 900 °C, which was due to the combination of the gas turbine with the heat collector and combustion chamber. Since these types of HA-CAES do not allocate TES, the output power of gas turbo-generator fluctuates with changes in solar radiation. Although the RTE and exergy efficiency are higher than the HT-CAES, this type of HA-CAES greatly depends on fossil fuel supplies, which leads to certain carbon emissions. Compared with the above HA-CAES, the HT-CAES with STS can provide a high-grade stable external heat source, which can greatly improve the stability and flexibility of HT-CAES.

### 4.2. Sensitivity Analysis

To investigate the effect of technique parameters on the performance of the HT-CAES, a sensitivity analysis was conducted. These parameters included inlet temperature and pressure of the compressors and turbine. The sensitivity analysis was conducted by varying one parameter, which caused affiliated parameters to vary correspondingly, while others were kept constant.

#### 4.2.1. Inlet Temperature of the Compressor

Figure 4a shows the effect of inlet temperature of the compressor on the W_COM_, W_TUR,_ and pressure inside the ASC (*P*_ASC_). The inlet temperature of the compressor slightly affects W_COM_ and *P*_ASC_, with W_TUR_ kept constant. With the increasing inlet temperature of the compressor, therefore, the ambient temperature also increased, which augmented the W_COM_ calculated by Equation (5). Thus, when the inlet temperature of the compressors increases from 15 °C to 35 °C, the energy consumption of W_COM_ increases by 8 kWh. Meanwhile, based on the ideal gas equation and constant volume of ASC, *P*_ASC_ increases by 0.15 MPa.

Figure 4b shows that the ESE, RTE, and η_ex_ all decline with increasing ambient temperature. As illustrated in Figure 4a, W_COM_ increases but W_TUR_ stays constant. Furthermore, the supplied heat load and heat energy provided by the STS remains constant. Therefore, when the ambient temperature increases from 15 to 35 °C, ESE, RTE, and η_ex_ all decrease by 1.1, 4, and 2, respectively. In general, a lower ambient temperature is more beneficial to system performance.

#### 4.2.2. Inlet Temperature of the Turbine

Figure 5a shows the effect of air turbine inlet temperature on W_COM_, W_TUR,_ and mass flow of VP-1. As revealed in Figure 5a, the increasing inlet air temperature of the turbine increases the power generation of the turbine system, owing to a greater decrease in the enthalpy of the air, which can be calculated by Equation (10). As the inlet temperature of the air turbine increases, the VP-1 mass flow also increases because the turbine needs more heat energy to enhance inlet air temperature. When the inlet temperature of the air turbine reaches 220 °C, the maximum W_TUR_ and mass flow of Therminol VP-1 reach 956 kWh and 2.7 kg/s, respectively. No compressor parameters were changed, so W_COM_ remained the same.

Figure 5b illustrates that the ESE, RTE, and η_ex_ increased with air turbine inlet temperature. As shown in Figure 5a, W_TUR_ increased while W_COM_ remained unchanged, which resulted in greater ESE according to Equation (21), while *Q_heat_* was constant due to the parameter being keep constant during the compression stage. Finally, by a comprehensive calculation using Equations (22) and (23), RTE and η_ex_ increase with the air turbine inlet temperature.

For example, when the air turbine inlet temperature increases from 180 °C to 220 °C under design inlet air pressure, ESE, RTE, and η_ex_ increase by 5.21%, 4.43%, and 4.38%, respectively. Therefore, all the efficiencies increase with the increasing air turbine inlet temperature. Moreover, the increments in the ESE, RTE, and η_ex_ are linear; the increasing rate of total output energy also increases with the increasing inlet air turbine pressure.

#### 4.2.3. Inlet Pressure of the Turbine

Figure 6a shows the variation of operation time and turbine power with inlet pressure of turbine. To eliminate throttling loss, we defined the minimum ASC pressure as equal to the inlet air turbine pressure. A greater turbine inlet pressure increases the output power of the turbine. However, when constrained by the ASC maximum pressure, increasing the ASC operation pressure leads to reduced system operation time. Meanwhile, the effect of ASC exerts minimum pressure on ESE, RTE, and η_ex_, as shown in Figure 6b. The decrease in the operation time reduced the amount of hot water and VP-1 consumption.

As shown in Figure 6b, when inlet pressure of turbine increases from 4.9 MPa to 8.8 MPa and the discharge duration time decreases from 6.9 h to 1.4 h, the output power increases by 29.7 kW. Correspondingly, ESE, RTE, and ηex increase by 8%, 5%, and 2.4%, respectively. Therefore, the combination of all these effects amplifies the ESE; RTE and ηex increase with the increase in inlet pressure of the turbine.

#### 4.2.4. Exhaust Pressure of the Compressor

Figure 7a shows the variation of the compressor’s power consumption *P_com_* and operation time with exhaust pressure of the compressor. Increasing the exhaust of the compressor means that the compression train needs to consume more energy to compress the air to a higher pressure. This also increases the system operation time. As shown in Figure 7a,b when the exhaust pressure of the compressor increases from 8 MPa to 12 MPa and the mass flow rate of air fluid is 2 t/h, the charge duration time increases from 1.4 h to 6.9 h, and the consumption power of compressors increases by 34.2 kW. Correspondingly, ESE, RTE, and η_ex_ decrease by 7.36%, 4.2%, and 4%, respectively. Therefore, ESE, RTE, and η_ex_ decrease with the increasing exhaust pressure of the compressor. Furthermore, the increasing mass flow rate of compressors leads to a gradual increase in compression power consumption.

### 4.3. Capability Analysis of CCHP

The HT-CAES that equips the solar thermal and refrigeration subsystems has a more flexible CCHP capacity. In order to fully excavate the capability of the HT-CAES, this section carries out the analysis of its multi-energy supply characteristics. In order to facilitate the analysis, the following assumptions are made:Thermal energy storage temperature in the hot tank is equal to supply heating and SAC;The temperature in the ASC is equal to the ambient temperature;The cooling water temperature and evaporation temperature of the SAC are constant;The PTC and SAC efficiency is constant;The inlet thermodynamic parameters of the TUR and COM stay constant;The minimum power supply time of the TUR is 0.5 h and at least 36% of the HTF is consumed.

The output electric power in the HT-CAES is shown in Section 3.3.2, and the compressed heat energy output for heating can be expressed as:(24)$$Q_{TES}=m_{O}\left(h_{O3}-h_{O2}\right).$$
where $m_{O}$ is the mass of VP-1, $h_{O2}\;$and $h_{O3}$ represent the specific enthalpy in the HOT and LOT, respectively.

The high-grade thermal energy of the HT-CAES is provided by VP-1 in the HOT, which can be used for heating/cooling and the output heating and cooling energy can be expressed respectively as:(25)$$Q_{HQ}=X\cdot Q_{TES}.$$
(26)$$Q_{CS}=Y\cdot COP_{ab}\cdot Q_{HQ}.$$
where X is the proportion of heat stored in the HOT for heating, *Y* is the proportion of cooling, and COP_ab_ is defined in Section 3.2. Table 7 lists the operation characteristics of the HT-CAES, with the heating proportion change from 0 to 64%. With the heating proportion gradually increased to 64%, the high-temperature heat transfer oil used for energy release and power generation is reduced, so the power supply time is shortened.

Under the premise of constant mass flow at the inlet of the turbine expander, the quality of high-pressure air consumed by the energy release power generation is reduced, which shows that the pressure range of the ASC is reduced from 8.0~3.0 MPa to 8.0~6.2 MPa, thus reducing the compression time and power consumption in the next cycle. Taking 24% of the heating proportion as an example, in the next compressed gas storage cycle, the air compressor only needs to pressurize the gas storage pipe from 4.2 MPa to 8.0 MPa, that is, the compression time is shortened, and the hot water recovered in the compression process is reduced.

Figure 8a shows the relationship between the output energy of the HT-CAES and the heating proportion X. With the change of heating proportion, in the next round cycle, the electric energy, heating energy and power supply energy consumed in energy storage decrease with the increase in heating proportion X and the external heating energy increases.

Further, to ensure the power generation capacity, the relationship between the heating/cooling capacity and the cooling ratio Y and the heating ratio X is analyzed, as shown in Figure 8b.

When the cooling ratio is fixed, the heating capacity increases linearly with heating ratio X. When the cooling ratio Y is 0 and the heating ratio X is 64%, the maximum heating capacity is 385.7 kwh, the regulation Y is 100%, and the maximum cooling capacity is 258.4 kwh. 

Figure 9 shows the relationship between the efficiency of the HT-CAES, CCHP and the heating ratio X and cooling ratio Y. When all the heat sources in the high-temperature oil tank are used for energy release power generation (i.e., x = 0), the system’s combined cooling and heating power efficiency is 80.4%; when the heating proportion is x = 64%, and the refrigeration proportion is Y = 0, and the system does not supply cooling to the outside, but the efficiency reaches the maximum value of 85%; when the heating proportion is X = 64%, and the refrigeration proportion is Y = 100%, and all the high-grade heat energy that can be deployed in the system is used for absorption refrigeration, under this energy supply mode, the lowest efficiency is 72.6%.

## 5. Conclusions

In this paper, a novel hybrid T-CAES system, i.e., a HT-CAES based on the utilization of solar thermal energy, is proposed and analyzed. Stable high-grade hot VP-1 stored in a HOT was used to provide cooling heating and power, which greatly improves the performance of the HT-CAES. To evaluate the system performance, energy and exergy analyses are conducted on the system. Finally, a sensitivity analysis and assessment capacity for CHP is conducted by the key parameters effected on performance of the HT-CAES. The main conclusions are summarized as follows:Charging and discharging times of the proposed system under design conditions are 5 h and 1.4 h, respectively. During these two modes, the system generates 498 kW h and consumes 940 kW h by the compressor and turbine. In addition, it generates 20 tons of hot water. In this situation, the ESE, RTE and exergy efficiency of the system are 53.6%, 73%, and 50.6%, respectively.A sensitivity analysis has indicated that the turbine inlet temperature and pressure are the critical parameters affecting the performance of the proposed HT-CAES system. When the increasing inlet temperature from 180 °C to 220 °C under design inlet air pressure, ESE, RTE, and η_ex_ increase by 5.21%, 4.43%, and 4.38%.Finally, the ratio of heating and cooling VP-1 was discussed to evaluate the CHP capacity of the proposed HT-CAES. When the heating proportion of VP-1 is x = 64%, and the refrigeration proportion is Y = 0, the RTE reaches the maximum value of 85%. Therefore, the proposed hybrid T-CAES can act as essential components in smart energy grid and cities, owing to the high efficiency and ability to accommodate renewables.

## Figures and Tables

**Figure 1 entropy-22-00764-f001:**
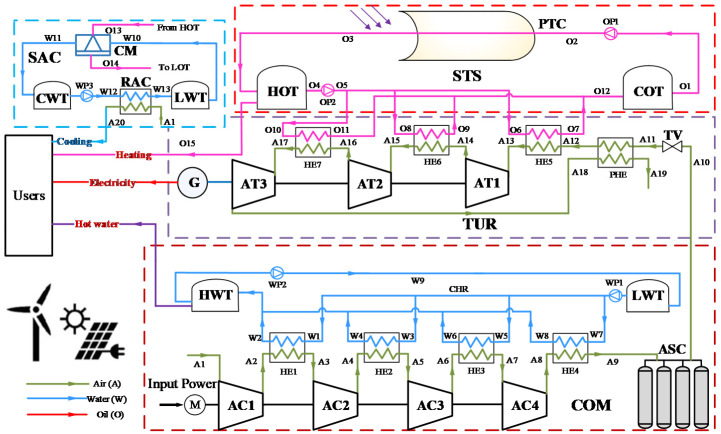
Schematic of the Hybrid trigenerative compressed air energy storage (HT-CAES) system.

**Figure 2 entropy-22-00764-f002:**
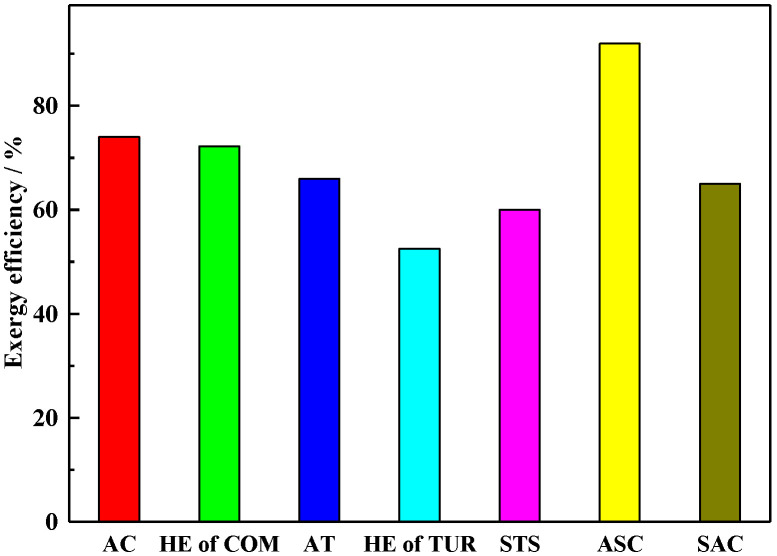
Exergy efficiencies of each subsystem.

**Figure 3 entropy-22-00764-f003:**
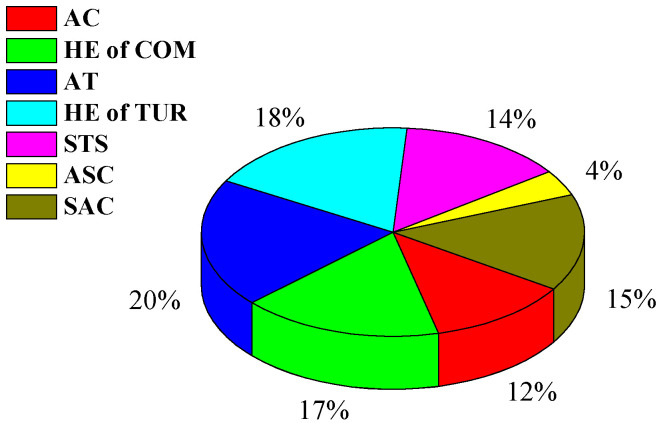
Exergy destruction of each subsystem.

**Figure 4 entropy-22-00764-f004:**
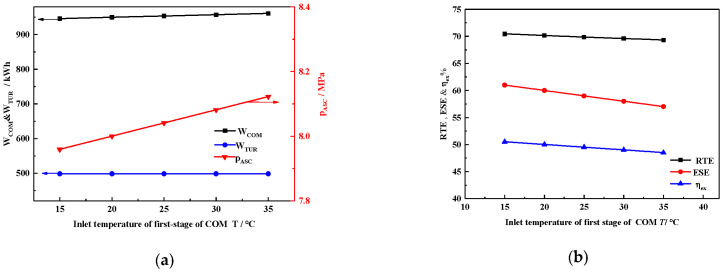
Effect of inlet temperature of compressors (**a**) on WCOM, WTUR, and PASC; (**b**) on ESE, RTE, and η_ex_.

**Figure 5 entropy-22-00764-f005:**
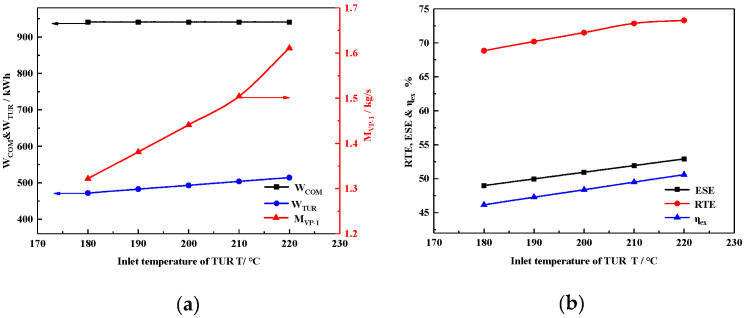
Effect of air turbine inlet temperature on (**a**) WCOM, WTUR, and VP-1 mass flow; (**b**) on ESE, RTE, and ηex.

**Figure 6 entropy-22-00764-f006:**
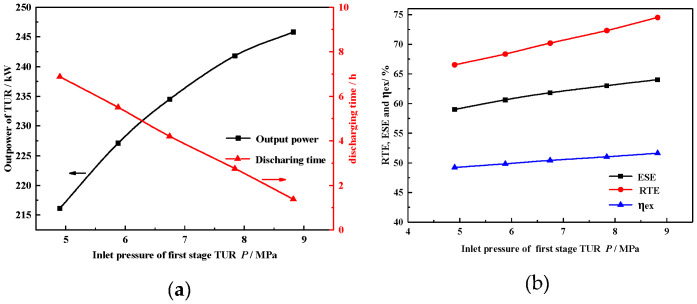
Effect of inlet pressure on (**a**) PTUR and operation time; (**b**) on ESE, RTE, and ηex.

**Figure 7 entropy-22-00764-f007:**
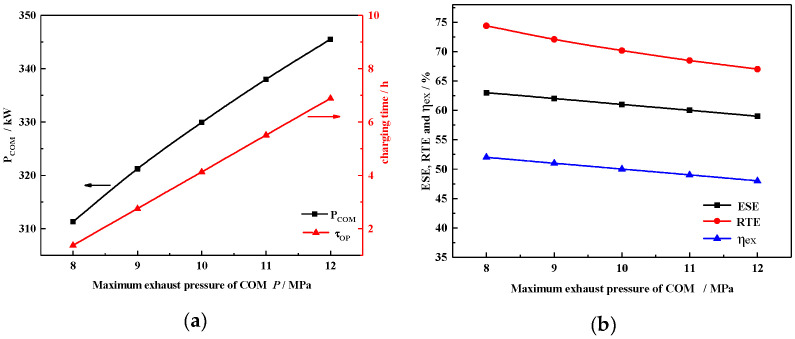
Effect of maximum exhaust pressure of COM (**a**) on PCOM and operation time; (**b**) on ESE, RTE, and ηex.

**Figure 8 entropy-22-00764-f008:**
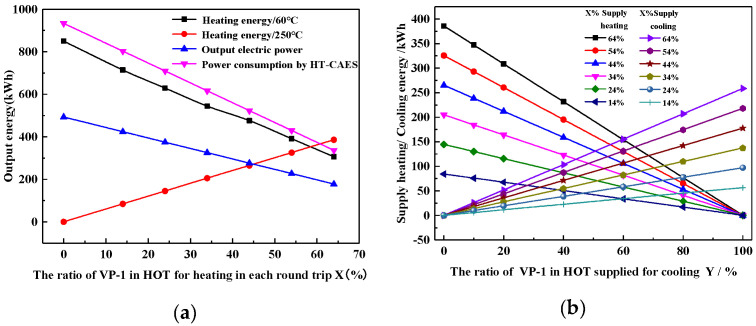
(**a**) Relationship between the output energy and the heating ratio X. (**b**) Relationship between heating/cooling and heating/cooling ratio X and Y.

**Figure 9 entropy-22-00764-f009:**
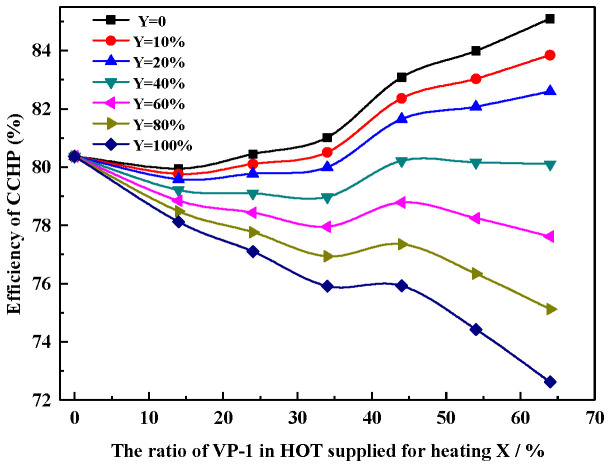
Relationship between the efficiency of the HT-CAES CCHP and the heating ratio X and cooling ratio Y.

**Table 1 entropy-22-00764-t001:** Expressions of input and output exergy of each subsystem.

Subsystem	Ex_in_	Ex_out_
AC	*W_COM_* + *Ex_A_*_3_ + *Ex_A_*_5_ + *Ex_A_*_7_	*Ex_A_*_2_ + *Ex_A_*_4_ + *Ex_A_*_6_ + *Ex_A_*_8_
HE of COM	*Ex_A_*_2_ + *Ex_A_*_4_ + *Ex_A_*_6_ + *Ex_A_*_8_ + *Ex_W_*_1_ + *Ex_W_*_3_ + *Ex_W_*_5_ + *Ex_W_*_7_	*Ex_A_*_3_ + *Ex_A_*_5_ + *Ex_A_*_7_ + *Ex_A_*_9_ + *Ex_W_*_2_ + *Ex_W_*_4_ + *Ex_W_*_6_ + *Ex_W_*_8_
STS	*Ex_Qu_*	*Ex_O_* _3_ − *Ex_O_* _2_
AT	*Ex_A_*_12_ + *Ex_A_*_14_ + *Ex_A_*_16_	*W_TUR_* + *Ex_A_*_13_ + *Ex_A_*_15_ + *Ex_A_*_17_
HE of TUR	*Ex_A_*_10_ + *Ex_A_*_17_ + *Ex_A_*_11_ + *Ex_A_*_13_ + *Ex_A_*_15_ + *Ex_O_*_5_	*Ex_A_*_11_ + *Ex_A_*_18_ + *Ex_A_*_12_ + *Ex_A_*_14_ + *Ex_A_*_16_ + *Ex_O_*_12_
Absorption Chiller	*Ex_O_*_13_ + *Ex_W_*_11_	*Ex_O_*_14_ + *Ex_W_*_12_
ASC	*Ex_A_* _9_	*Ex_A_* _10_

**Table 2 entropy-22-00764-t002:** Design parameters of the Hybrid trigenerative CAES (HT-CAES) system.

Parameters	Units	Values	Parameters	Units	Values
COM			TUR		
Ambient pressure	MPa	0.1	Inlet pressure of air turbine	MPa	3
Ambient temperature	°C	20	Inlet temperature of air turbine	°C	200
Compression stage (*i*)	/	4	Expansion stage (*j*)	/	3
Isentropic efficiency of compressor	%	90	Isentropic efficiency of turbine	%	85
Compression ratio	/	3	Expansion ratio	/	2.65
τ_ch_	h	5	τ_dch_	h	1.4
Mass flow rate of compressor	kg/s	0.33	Mass flow rate of turbine	kg/s	1.17
Temperature of recovery water	°C	30	Range of pressure with ASC	MPa	3~8
Volume of air store chamber	m^3^	100	Efficiency of generator	%	94.8
Temperature of hot water tank	°C	60	SAC		
/	/	/	Cooling water supplying duration	h	5
Temperature of low water tank	°C	20	COP of absorption chiller	/	0.67
Hot water supplying duration	h	8	Mass flow rate of cooling water	kg/s	0.63
STS					
Direct normal irradiance	W/m^2^	841.1	Range of temperature with hot oil tank	°C	250~252
Mass flow rate of VP-1 (charging process)	kg/s	0.336	Solar thermal storage duration	h	6
Efficiency of heat collection	%	63.8	Mass flow rate of VP-1 (discharging process)	kg/s	1.44
Reflectivity of collector mirrors	%	94	Collector area	m^2^	405.8

**Table 3 entropy-22-00764-t003:** Simulation results of the HT-CAES system under typical operational conditions.

Parameters	Unit	Values
Compressor power consumption	kW	190
Air turbine electricity generation	kW	352
Collection power of PTC	kW	100.6
Production the mass of cooling water	ton	15
Production the mass of hot water	ton	20
Electricity storage efficiency	%	53.6
Round-trip efficiency	%	73
Exergy efficiency	%	50.6

**Table 4 entropy-22-00764-t004:** Thermodynamic parameters of air stream.

Stream	T (°C)	P (MPa)	h (kJ/kg)	s (kJ/kg·K)	Ex (kJ/kg)	m (kg/s)
A1	20	0.101	419.41	3.8644	0	0.33
A2	143.7	0.309	544.15	3.8983	114.81	0.33
A3	45	0.303	444.17	3.6302	93.403	0.33
A4	178.8	0.924	579.49	3.6648	218.6	0.33
A5	45	0.906	442.98	3.3124	185.4	0.33
A6	177.3	2.729	576.64	3.3461	309.18	0.33
A7	45	2.675	439.58	2.9916	276.05	0.33
A8	179.6	8.16	575.76	3.0246	402.53	0.33
A9	45	8	430.23	2.6485	367.26	0.33
A10	19.9	2.903	412.82	2.8806	281.82	1.17
A11	65	2.903	460.02	3.0304	285.1	1.17
A12	200	2.846	600.13	3.3848	321.32	1.17
A13	90	0.949	488.73	3.4336	195.62	1.17
A14	200	0.93	601.28	3.71	227.12	1.17
A15	90.2	0.31	489.85	3.758	101.65	1.17
A16	200	0.304	601.68	4.0324	133.02	1.17
A17	90.3	0.101	490.26	4.081	7.3474	1.17
A18	34.5	0.1	308.59	6.89	0.35	1.17

**Table 5 entropy-22-00764-t005:** Thermodynamic parameters of Therminol VP-1 and water stream.

Stream	T (°C)	P (Mpa)	h (kJ/kg)	s (kJ/kg·K)	Ex (kJ/kg)	m (kg/s)
O1	100	0.101	579.85	−7.2	10.21	0.34
O2	100.3	0.436	580.29	−2.77	14.90	0.34
O3	251	0.101	879.18	−1.48	273.06	0.34
O4	250	0.1014	877	−2.03	144.64	1.44
O5	250	0.103	877	−1.50	268.91	1.44
O6	250	0.103	877	−1.50	268.91	0.55
O7	100	0.101	579.85	−2.76	10.54	0.55
O8	250	0.103	877	−1.50	268.91	0.45
O9	100	0.101	579.85	−2.65	21.99	0.45
O10	250	0.103	877	−1.50	268.91	0.44
O11	100	0.101	579.85	−2.64	22.22	0.44
O12	100	0.101	579.85	−2.69	17.87	1.44
O13	250	0.12	375.20	−1.02	144.64	0.9
O14	150	0.101	14.30	−2.06	14.90	0.9
W1	20	0.103	84.01	0.30	0.00	0.2
W2	60	0.101	251.25	0.83	10.47	0.2
W3	20	0.103	84.01	0.30	0.00	0.27
W4	60	0.101	251.25	0.83	10.47	0.27
W5	20	0.103	84.01	0.30	0.00	0.27
W6	60	0.101	251.25	0.83	10.47	0.27
W7	20	0.103	84.01	0.30	0.00	0.28
W8	60	0.101	251.25	0.83	10.47	0.28
W9	60	0.403	251.50	0.83	10.77	1.02
W10	20	0.403	84.29	0.30	0.30	1.02
W11	60	0.130	251.27	0.83	10.50	0.63
W12	30	0.101	125.82	0.44	0.70	0.63
W13	4	0.2	17.01	0.061	2.0	0.4
W14	19.6	0.2	82.43	0.29	0.098	0.4

**Table 6 entropy-22-00764-t006:** The comparison of performance of the ST-CAES to the hybrid A-CAES.

Parameters	Unit	HT-CAES	Hybrid A-CAES
Compressor power consumption	kWh	925	152.1
Charging time	h	5	6.59
Discharging time	h	1.4	5.13
With/without TES	/	Yes	no
Expansion train working period		load peak hours	Irradiation peak hours
Inlet temperature of air turbine	°C	200	900
Air turbine electricity generation	kWh	385.2	228.54
Round trip efficiency	%	73	76.5
Exergy efficiency	%	50.6	53.4

**Table 7 entropy-22-00764-t007:** The relationship between the operating characteristics and the heating proportion.

Heating Ratio X%	VP-1 Mass for Generating Power T	Regenerated Hot Water T	The Range Pressure in ASC MPa	$\tau_{ch}$ h	$\tau_{dch}$ h
0	7.2576	18.3	8.0~3.0	5.0	1.4
14	6.24154	15.4	8.0~3.7	4.2	1.2
24	5.51578	13.5	8.0~4.2	3.7	1.0
34	4.79002	11.7	8.0~4.7	3.2	0.9
44	4.06426	10.2	8.0~5.2	2.7	0.7
54	3.3385	8.4	8.0~5.7	2.2	0.6
64	2.61274	6.6	8.0~6.2	1.7	0.5

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
