# Peer review of "Thermodynamic Analysis of a Hybrid Trigenerative Compressed Air Energy Storage System with Solar Thermal Energy"

_entropy, 2020, doi:10.3390/e22070764_

Round 1

Reviewer 1 Report

The authors have presented a numerical study of HT-CAES system. While CAES has been studied rigorously in the literature, the integration of CAES with other applications for better heat integration has not been explored much, and the results in this paper can be of interest to the CAES community for making the system more efficient. I have some questions and comments to improve the paper:

  1. In line 90, I don’t know what the authors meant by ‘high grade solar thermal energy.’ Is it high temperature? And why does the system supply ‘solar thermal energy’ in the discharging state?
  2. In this paper, I couldn’t find how the system was modelled, except for some equations defined part of the process. I will suggest the authors provide details about the system modelling for reproducibility. To name a few: did the authors use any software to build the model? How did they derive the thermal properties of other working fluids other than gases (assumption 2)? How did the authors verify the results of the model? How is the solar absorption system modelled? And so on…
  3. I don’t agree with the authors on the definition of RTE. As the authors propose to analyze the exergy efficiency, I think it should be understandable that it is not a good idea to directly add heat and electricity energy together, without considering the quality of the energy. That’s to say, the RTE should be the round trip exergy efficiency, rather than the one defined in equation (22). And it is rather confusing to present the efficiency defined in (22), so I suggest the authors not to use it, unless they have other reasons.
  4. In line 220, how did the authors decide on the duration of charging and discharging?
  5. For table 2, please categorize the properties into different systems described earlier.
  6. In section 4.2.*, in the subtitles, they should be the inlet temperature of compressors or turbines, rather than the systems. Also, for the figures in the same sub-section, please put them into one title, and list them as a and b, rather than using separate figure titles. It is not easy to read so many figures.
  7. In the paragraph starting at line 270, the authors compared HT-CAES with HA-CAES, and they found that the HT-CAES is not as good as HA-CAES in terms of efficiency. But then they said that HA-CAES emits CO2. How about comparing with the systems not using fossil fuel? And how about HA-CAES with biomass generated methane, which is CO2 neutral? I think the authors have to re-consider the comparisons.
  8. T-CAES has not been defined explicitly in their paper. Please add that into the revised manuscript.

Reviewer 2 Report

Line 49: Now the plant has 321 MW. A specification should be inserted here due to this situation.

Line 125: The range of the air temperatures is quite large. Keeping the specific heat constant should be much more sustained here, due to the fact that you are producing conclusions with values with 2 decimals. 

Equation 2: On the left term mass flow rate should be considered. The dots over the letters are missing.

Line 142: The therminol VP-1 is very important in your paper, but it is missing in the general description of the installation. I think a certain gain in clearance of the description should be gained if you are inserting this VP-1 (why 1?) among the lines 98-117.

Line 152: You should make a review of all the symbols of mass flow rate.

Line 179: You should insert the assumption concerning the Joule-Thomson effect in the line 131. It is a very strong assumption.

Line 191: The same as the line 152.

Reviewer 3 Report

  1. Abstract; electricity storage eff is higher than round-trip efficiency, which seems not normal.

Round-trip should include 2 way energy storage, which is supposed to be less than single storage efficiency. Please check. (Table 3 as well). In fact, you can define electricity storage separately, and energy storage (including thermal) separately and then total energy storage (including both thermal and electrical). In this way, it will be more clear to readers.

  1. Is ideal gas assumption reasonable for your conditions? You can show a simple comparison in the paper.
  2. What was the main reason of having different number of compression and expansion stages?
  3. Table 2; generator eff. was it assumed or calculated? What do you mean by the adiabatic eff. or turbine and compressor? Do you mean the isentropic eff? or first law eff? Please write clearly.
  4. 2. Is it ex-COP for absorption chiller? Write clearly.
  5. 6.; does it show that there is no relation between compressor exit temp and turbine inlet temp? If yes, is this because of ASC?
  6. How to set the discharging time if there is a specific load to be run specific hours? Is there any methodology?
  7. What does Fig. 14 exactly imply? Can the operator of the system change the ratio of VO-1 supplied for heating and adjust the load and supply?
  8. Conclusions; please include some main findings of your study numerically.

Round 2

Reviewer 1 Report

The authors have modified the paper based on reviewer's comments. And the quality has been improved a lot. I recommend the paper to be accepted.